# A genome-wide view of the de-differentiation of central nervous system endothelial cells in culture

**Mark F Sabbagh[1,2], Jeremy Nathans[1,2,3,4]\***

[1]Department of Molecular Biology and Genetics, Johns Hopkins University School of Medicine, Baltimore, United States; [2]Department of Neuroscience, Johns Hopkins University School of Medicine, Baltimore, United States; [3]Department of Ophthalmology, Johns Hopkins University School of Medicine, Baltimore, United States; [4]Howard Hughes Medical Institute, Johns Hopkins University School of Medicine, Baltimore, United States

**Abstract** Vascular endothelial cells (ECs) derived from the central nervous system (CNS) variably lose their unique barrier properties during in vitro culture, hindering the development of robust assays for blood-brain barrier (BBB) function, including drug permeability and extrusion assays. In previous work (Sabbagh et al., 2018) we characterized transcriptional and accessible chromatin landscapes of acutely isolated mouse CNS ECs. In this report, we compare transcriptional and accessible chromatin landscapes of acutely isolated mouse CNS ECs versus mouse CNS ECs in short-term in vitro culture. We observe that standard culture conditions are associated with a rapid and selective loss of BBB transcripts and chromatin features, as well as a greatly reduced level of beta-catenin signaling. Interestingly, forced expression of a stabilized derivative of beta-catenin, which in vivo leads to a partial conversion of non-BBB CNS ECs to a BBB-like state, has little or no effect on gene expression or chromatin accessibility in vitro.

\*For correspondence:
jnathans@jhmi.edu

## Introduction

Vascular endothelial cells (ECs) develop unique cellular and molecular properties to meet the specialized needs of target tissues (*Augustin and Koh, 2017*; *Potente and Mäkinen, 2017*). In the central nervous system (CNS) vasculature, the object of study here, ECs acquire a set of properties that collectively form the blood-brain barrier (BBB) or its retinal equivalent, the blood-retina barrier (BRB). This unique EC phenotype arises from a genetic regulatory program that (1) suppresses the production of fenestrae- and caveolae-associated components, thereby reducing intra-endothelial diffusion and transcytosis, respectively, and (2) enhances the production of tight junction proteins that block intercellular diffusion, transporters that carry small molecules such as glucose and amino acids from the blood to the brain parenchyma, and efflux pumps that extrude toxic molecules from the brain parenchyma (*Zhao et al., 2015*).

Numerous studies have found that cultured ECs have a propensity to lose their tissue-specific properties (*DeLeve et al., 2004*; *Durr et al., 2004*; *Lacorre et al., 2004*; *Sellaro et al., 2007*; *March et al., 2009*; *Burridge and Friedman, 2010*; *Géraud et al., 2010*; *Kim and Rajagopalan, 2010*; *Tasnim and Zink, 2012*; *Urich et al., 2012*; *Uhlig et al., 2014*; *Zhao et al., 2014*; reviewed in *Helms et al., 2016*). In the case of CNS ECs, overcoming this propensity is of both theoretical and practical interest. An EC culture system that could faithfully reproduce the small molecule permeability, transport, and extrusion properties of the BBB would permit high throughput in vitro screening of drug candidates to predict their brain penetration (*Helms et al., 2016*). Loss of the CNS EC-specific expression profile by cultured brain ECs could result from a loss of brain-specific signaling,

including beta-catenin signaling (also called canonical Wnt signaling). Conditional deletion of beta-catenin signaling components in ECs in vivo leads to loss of BBB-specific gene expression (*Liebner et al., 2008*; *Stenman et al., 2008*; *Daneman et al., 2009*; *Wang et al., 2012*; *Zhou et al., 2014*; *Cho et al., 2017*; *Hupe et al., 2017*). Conversely, elevating beta-catenin signaling in ECs in vivo leads to a partial conversion of highly permeable ECs in the circumventricular organs (CVOs) and choroid plexus to a BBB state (*Benz et al., 2019*; *Wang et al., 2019*).

Cultured ECs have traditionally been characterized by assessing their morphological and functional properties and by quantifying the abundances of a small number of marker proteins or transcripts. With the advent of NextGen sequencing technologies, it is now possible to profile the transcriptome and chromatin state on a genome-wide scale to obtain a comprehensive and quantitative view of cellular identity (e.g., *Gosselin et al., 2014*; *Lavin et al., 2014*; *Sabbagh et al., 2018*). In the present study, we have characterized the effects of short-term in vitro culture on CNS EC identity at the level of the transcriptional and accessible chromatin landscapes. This analysis reveals both a rapid loss of BBB properties and resistance to the effects of beta-catenin stabilization during in vitro culture.

## Results

### Rapid loss of the CNS EC transcriptional signature upon culturing of primary brain ECs

To acquire an unbiased, genome-wide view of the effect of in vitro culture on the transcriptional and chromatin landscapes of CNS ECs, we grew primary cultures of young adult (8–16 week) brain ECs in standard EC culture medium for eight days (see Materials and methods) and then performed RNA-seq and ATAC-seq on the resulting cells. To selectively culture brain ECs, we took advantage of the expression of P-glycoprotein – an efflux pump with broad substrate specificity – by CNS ECs but not by neurons, glia, or pericytes. Following dissociation of brain cells and enrichment for vascular fragments, the mixed culture was treated with puromycin for 2–2.5 days, which kills non-ECs and selectively spares ECs, a widely used method for preparing pure cultures of acutely isolated CNS ECs (*Figure 1A*; *Welser-Alves et al., 2014*; *Assmann et al., 2017*; reviewed in *Helms et al., 2016*). ECs were cultured on collagen-coated wells for eight days in 5% serum with supplementary growth factors (see Materials and methods). Immunostaining of the resulting EC cultures revealed uniform expression of the pan-EC marker CD31 (PECAM1) across the monolayer (*Figure 1—figure supplement 1A*).

We repeated the culture experiment three times, with cells from independent wells in each experiment serving as biological replicates for a total of six replicates for both RNA-seq and ATAC-seq (*Figure 1—figure supplement 1B*). Comparing the RNA-seq data for all protein-coding genes among the three independent experiments, the Pearson correlation ranged from 0.93 to 0.96 (*Figure 1—figure supplement 1C*). An analogous comparison between two replicates of acutely isolated brain ECs (purified by FACS from the cerebrum and cerebellum of *Tie2-GFP* [also known as *Tek-GFP*] mice) gave a Pearson correlation of 0.93. [The cerebrum EC data is new to this study; the cerebellum EC data were published previously (*Wang et al., 2019*). For the present study, we combined the two EC data sets (cerebrum and cerebellum) to generate what we call 'adult brain ECs'.] Comparisons within the paired replicates in the three cultured EC experiments gave a Pearson correlation of 0.99–1.00. In comparing acutely isolated versus cultured brain ECs, 3585 protein-coding transcripts met our criteria for differential expression (see Materials and methods): 1542 with a > 2 fold increase in abundance in acutely isolated ECs and 2043 with a > 2 fold increase in abundance in cultured ECs. Since some of the transcripts in the acutely isolated brain EC sample could have come from contaminating brain parenchymal cells, we restricted our analysis of transcripts enriched in this sample to a set of 1352 transcripts that were enriched >2 fold in GFP+ cells (ECs) over GFP- cells (non-ECs, that is parenchymal cells) (see Materials and methods). Among these 1352 transcripts, 581 exhibited a > 2 fold increase in abundance in acutely isolated ECs compared to cultured ECs. For a list of these genes, see *Supplementary file 1*.

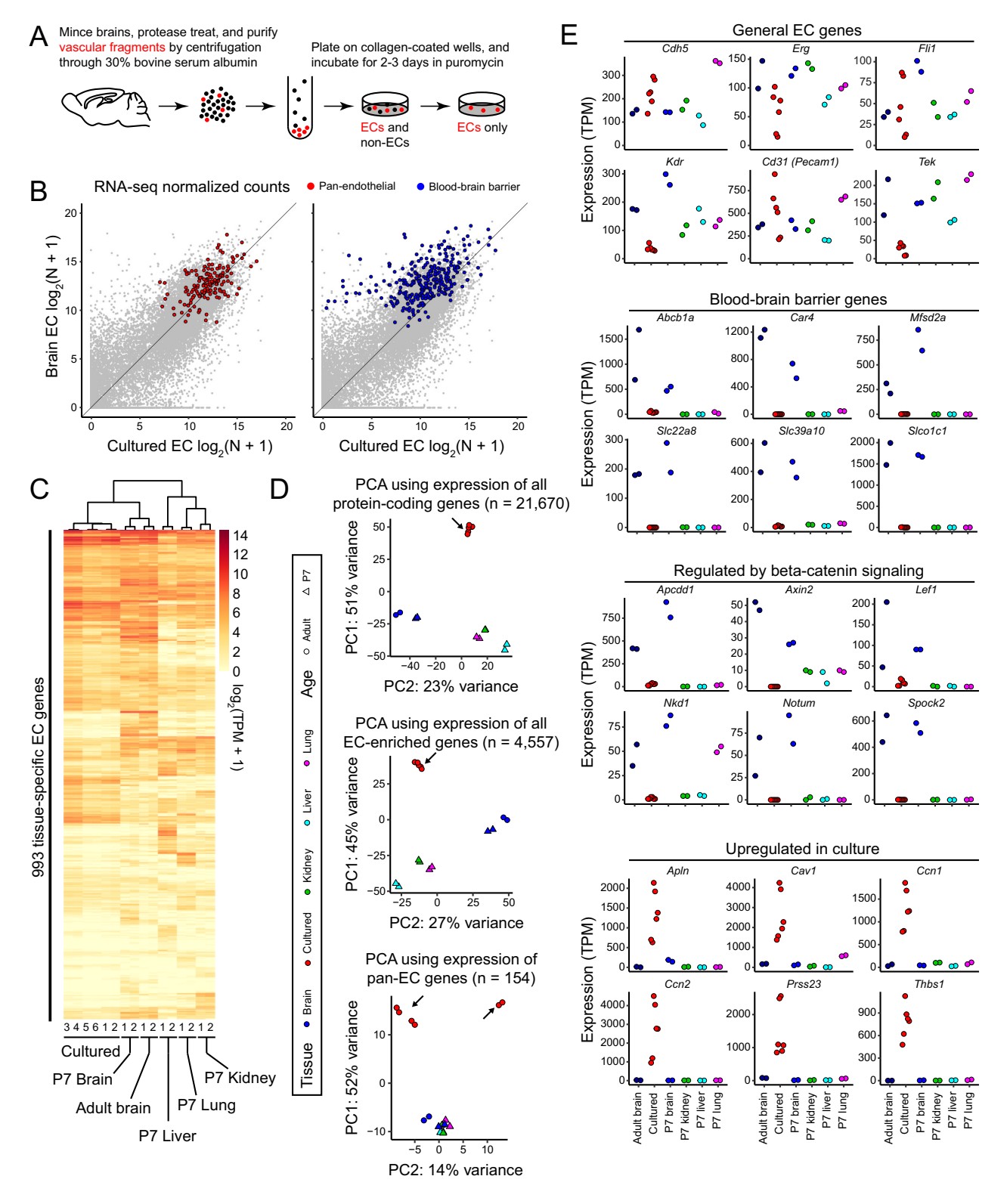

**Figure 1.** RNA-seq reveals a rapid loss of the brain EC transcriptional signature during in vitro cell culture. (**A**) Schematic outline of the isolation, purification, and culture of brain ECs. (**B**) Scatter plots comparing cross-sample normalized RNA-seq read counts for all protein-coding genes (gray symbols) between acutely isolated adult brain ECs (y-axis; average of two independent samples) and primary brain ECs cultured for 8 days (x-axis; average of six independent samples). The left plot highlights (in red) EC-enriched genes that are expressed at similar levels in adult brain, P7 brain, P7

*Figure 1 continued on next page*

*Figure 1 continued*

liver, P7 lung, and P7 kidney ECs. The right plot highlights (in blue) genes associated with the BBB (i.e., that are >2 fold enriched in brain ECs relative to liver, lung, and kidney ECs). The majority of BBB genes show reduced expression in culture. (C) Heatmap depicting $\log_2$ transformed transcript abundances from cultured brain ECs, and from acutely isolated P7 and adult brain ECs, and P7 liver, lung, and kidney ECs for 993 tissue-specific EC genes. (D) Principal component analysis (PCA) based on the abundances of all protein-coding transcripts (top), EC-enriched transcripts (center), and pan-EC transcripts (i.e. transcripts common to adult brain, P7 brain, P7 liver, P7 lung, and P7 kidney ECs; bottom). The symbols for each sample represent biological replicates. Arrows indicate cultured EC data points. In the PCA in the bottom panel, the two samples in the upper right were replicates isolated by Method B, and their RNA-seq libraries were generated and sequenced by a different facility compared to the other four samples, which were isolated by method A (see Materials and methods). (E) Transcript abundances for six genes from each of the four indicated categories of transcripts. Labels at the bottom of the figure apply to all panels above.

The online version of this article includes the following figure supplement(s) for figure 1:

**Figure supplement 1.** Characterization of primary brain EC cultures.
**Figure supplement 2.** Further analysis of the brain EC transcriptional signature during in vitro cell culture.
**Figure supplement 3.** Analysis of transcripts coding for ABC transporters and proteins associated with Clathrin- or Caveolin-mediated transcytosis.

To more precisely define transcriptional changes in acutely isolated versus cultured brain ECs, we separately assessed abundance changes among pan-EC transcripts and among BBB transcripts (*Supplementary file 1*). We define pan-EC transcripts based on a > 2 fold increase in abundance in ECs relative to parenchymal cells in adult brain and in four different EC subtypes analyzed at postnatal day (P)7 (brain, liver, lung, and kidney; *Sabbagh et al., 2018*) and on a lack of differential abundance (<2 fold) among these EC subtypes. We define BBB transcripts based on a > 2 fold increase in abundance in adult brain ECs compared to P7 liver, lung, and kidney ECs. Scatter plots comparing between-sample normalized RNA-seq read counts for acutely isolated versus cultured adult brain ECs show that, on average, pan-EC transcripts are present at similar levels, whereas most BBB transcripts show reduced abundance in the cultured brain ECs [*Figure 1B*]. This analysis indicates that, with eight days in culture, CNS ECs retain the core EC transcriptional program but begin losing the CNS-specific transcriptional program.

We next assessed the effect of in vitro culture on the expression of 993 genes that exhibit enriched expression in one or another of the four EC subtypes profiled at P7 (brain, liver, lung, and kidney) or in adult brain ('EC tissue-specific genes'; *Supplementary file 1*). A dendrogram of the relatedness of the different EC samples shows that cultured adult brain ECs are more closely related to P7 and adult brain ECs than to P7 liver, lung, or kidney ECs (*Figure 1C*). The same pattern is observed if the analysis is expanded to all 4,557 EC-enriched genes (*Figure 1—figure supplement 2A*). Principal component analysis (PCA) of transcript abundances for all protein-coding genes shows substantial divergence of the cultured brain EC transcriptome compared to the in vivo brain, liver, lung, and kidney EC transcriptomes (upper panel of *Figure 1D*). This PCA also shows the close similarity of the P7 and adult in vivo brain EC transcriptomes. PCA of transcript abundances for all EC-enriched genes (middle panel of *Figure 1D*) and for pan-EC genes (bottom panel of *Figure 1D*) shows a similar divergence of the cultured brain EC transcriptome compared to the transcriptional profiles of the in vivo EC subtypes. In the bottom panel of *Figure 1D*, the upper right pair of cultured EC samples used a cell isolation protocol and RNA-seq pipeline that differed from those used for the four samples in the upper left, as described in the *Figure 1* legend and Materials and methods section.

In previous work (*Sabbagh et al., 2018*), single cell RNA-seq (scRNA-seq) of P7 brain ECs was used to define six major cell clusters: arterial, venous, mitotic, tip cell, and two capillary clusters, capillary-A and capillary-V; the latter two share some features with arterial and venous gene expression patterns, respectively. Using the top 500 transcripts enriched in each of the six scRNA-seq clusters, we compared expression levels between acutely isolated and cultured brain ECs (*Figure 1—figure supplement 1D*). The cultured brain ECs showed reduced expression for some of the genes associated with the two capillary clusters (*Figure 1—figure supplement 1D*, 'Capillary-A' and 'Capillary-V' panels), consistent with the BBB program manifesting primarily at the capillary level. The cultured brain ECs also show an increase in expression of several genes associated with the mitotic cluster, consistent with the proliferation of ECs in culture (red arrow in *Figure 1—figure supplement 1D*).

To further explore the effect of in vitro culture on brain ECs, we analyzed the expression of different EC gene sets. As noted above, cultured brain ECs maintain relatively high-level expression of generic EC transcripts, although with some variability from sample-to-sample, and they exhibit reduced expression of BBB genes (181/279 are reduced >2 fold; *Figure 1B* and *Figure 1E*). Genes known to be activated in response to beta-catenin signaling or whose protein products regulate beta-catenin signaling (*Behrens et al., 1998*; *Shimomura et al., 2010*; *Chang and Sun, 2014*; *Kakugawa et al., 2015*; *Mazzoni et al., 2017*) showed little or no expression in cultured brain ECs (*Figure 1E*), implying a lack of beta-catenin signaling under these culture conditions. We note that five of these genes – *Apcdd1, Axin2, Lef1, Notum,* and *Spock2* – also meet the criteria for inclusion as 'BBB genes', in accordance with the established role of beta-catenin signaling in CNS ECs. The abundances of six transcripts that are increased in cultured brain ECs compared to acutely isolated brain ECs are plotted in the last panel of *Figure 1E*.

We further examined the effect of in vitro culture on four categories of transcripts coding for BBB-associated proteins: (1) tight junction (TJ) proteins, (2) solute carrier family transporters, (3) ABC transporters, and (4) transcytosis-associated proteins (*Tietz and Engelhardt, 2015*; *Sabbagh et al., 2018*; *Ayloo and Gu, 2019*). These data are shown in *Figure 1—figure supplements 2B* and *3*, which also includes the transcriptome results for brain ECs cultured from mice with stabilized beta-catenin, as described below. Among the more abundant transcripts coding for TJ proteins there is a mixture of responses to in vitro culture: *Cldn5* transcripts show little change, *Lsr* and *Ocln* transcripts are reduced ~2 fold, *Jam2* and *Tjp1* transcripts are reduced ~5 fold, and *F11r* and *Tjp2* are elevated 2–4-fold in a subset of the cultured EC samples. Among transcripts coding for solute carrier family transporters, in vitro culture had either no effect (e.g., *Slc16a9, Slc1a1,* and *Slc46a3*) or it reduced expression, in several cases to a nearly undetectable levels (e.g., *Slc22a8, Slc38a3, Slc39a10,* and *Slco1c1*). The levels of multiple transcripts coding for ABC transporters are either unaffected (e.g., *Abcc4* and *Abcg2*) or elevated (e.g., *Abca3, Abcb1b, Abcc1,* and *Abce1*) by in vitro culture; the one exception is the most abundant isoform of P-glycoprotein (*Abcb1a*), which is greatly reduced by in vitro culture. Among transcytosis-associated transcripts, in vitro culture causes a dramatic reduction in the abundance of transcripts coding for the transcytosis regulator Mfsd2a (*Figure 1E*; *Ben-Zvi et al., 2014*) and concomitant increases in the abundances of transcripts coding for Caveolin-1, Caveolin-2, Cavin-1, Cavin-3, and Ehd2 (components of the Caveolin-mediated transcytotic pathway). Among 16 transcripts coding for proteins associated with Clathrin-mediated transcytosis, only *Clta* and *Cltb* show consistent elevation with in vitro culture.

In summary, the broad reduction in the expression of beta-catenin-responsive and BBB genes implies a reduction in beta-catenin signaling and the BBB gene expression program in brain ECs cultured in vitro.

## Changes in accessible chromatin in cultured brain ECs reflect a loss of beta-catenin signaling

The identification of gene expression changes in cultured brain ECs implies corresponding changes in the chromatin landscape. Comparisons of accessible chromatin in acutely isolated versus cultured brain ECs show (1) loss of ATAC-seq peaks near multiple brain EC-specific genes that lose expression in cultured ECs, such as *Slco1c1* and *Mfsd2a*, and (2) gain of ATAC-seq peaks near multiple genes that are induced in cultured ECs, such as *Thbs1* and *Cyr61* (also known as *Ccn1*) (*Figure 2A*; the histograms in the genome browser images represent averages of the independent replicates). PCA of ATAC-seq read density at a consensus set of called ATAC-seq peaks indicates that the accessible chromatin landscapes of acutely isolated adult and P7 brain ECs are as different from cultured brain ECs as the latter is from peripheral ECs (*Figure 2B*). PCA also indicates that acutely isolated adult and P7 brain ECs have very similar accessible chromatin landscapes. Among 102,016 ATAC-seq peaks identified in the acutely isolated and cultured brain EC datasets, 16,760 were lost and 20,654 were gained as a result of in vitro culture (*Figure 2C*; *Supplementary file 2*).

Transcription factor (TF) motif enrichment analysis of ATAC-seq peaks revealed two distinctive trends: (1) accessible regions gained during in vitro culture were strongly enriched for motifs of the activator protein-1 (AP-1) complex (FOS, JUN, and ATF) and TEAD families, and (2) accessible regions lost during in vitro culture were strongly enriched for TCF/LEF, SOX, and SP/KLF motifs (*Figure 2D and E*). The AP1 and TEAD signatures in regions of gained open chromatin likely reflect cell proliferation and/or a stress response to the in vitro environment. The TCF/LEF signature in

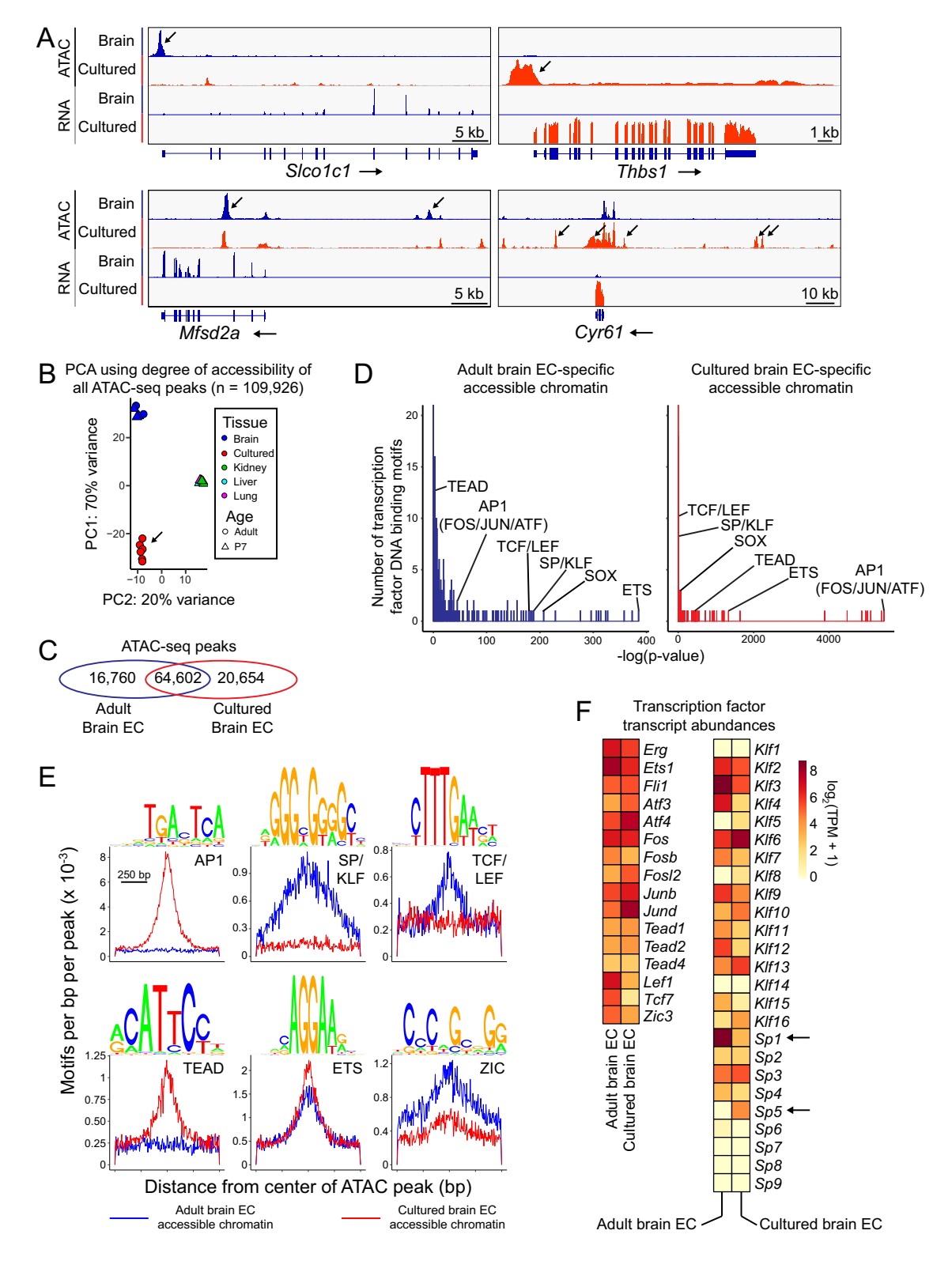

**Figure 2.** Changes in accessible chromatin, motif enrichment, and TF expression in primary brain ECs in culture. (**A**) Genome browser images showing ATAC-seq reads (top) and RNA-seq reads (bottom) at or near *Slco1c1* and *Mfsd2a*, BBB genes with reduced expression in cultured adult brain ECs (left panels), and *Thbs1* and *Cyr61* (*Ccn1*), genes with enhanced expression in cultured adult brain ECs (right panels). Increases in transcript abundance are accompanied by increases in accessible chromatin near each gene (black arrows). Read counts are averaged over the independent replicates

*Figure 2 continued on next page*

Figure 2 continued

(**Figure 1—figure supplement 1**). (B) PCA of ATAC-seq read density at all called ATAC-seq peaks in independent biological replicates from cultured adult brain ECs, acutely isolated adult and P7 brain ECs, and P7 liver, lung, and kidney ECs. Liver, lung, and kidney samples cluster at center right. The arrow points to cultured ECs. (C) Venn diagram summarizing the number of shared and distinct ATAC-seq peaks between acutely isolated and cultured adult brain ECs. (D) TF motif enrichment in ATAC-seq peaks that is specific to either acutely isolated adult brain ECs (left) or cultured adult brain ECs (right). Histograms of the -log$_{10}$(p-value) for 414 TF DNA binding motifs that were tested for enrichment. The x-axis bin size is 1. The y-axis has been truncated. The vast majority of tested motifs have high p-values (i.e., low statistical significance) and thus are found in the first few bins. The statistically most significant TF motif families are labeled. Most TF families are represented by multiple closely related motifs, and only the motif with the lowest p-value for each family is labeled. The number of ATAC-seq peaks analyzed in (D) and (E) corresponds to the values shown in the Venn diagram in (C). (E) Enriched TF motifs identified by HOMER. The frequency of the indicated motifs is plotted as a function of distance from the center of the ATAC-seq peaks (from either acutely isolated or cultured adult brain ECs). Shown above each individual plot is the position weight matrix (PWM) of the enriched nucleotide sequence. The TF family that most closely matches the motif is indicated below the PWM. (F) Heatmap showing log$_2$ transformed TPMs for acutely isolated and cultured adult brain ECs for transcripts coding for a subset of TFs with the motifs shown in (E).

regions of open chromatin lost during in vitro culture likely reflects the loss of beta-catenin signaling under standard EC culture conditions. The SP/KLF signature in regions of open chromatin that are lost during in vitro culture could reflect the lack of laminar flow in our culture conditions, as both *Klf2* and *Klf4* expression in ECs is induced by laminar flow (**Atkins and Jain, 2007**). Both sets of accessible regions show similar enrichment for motifs of the ETS family of TFs, master regulators of EC development and function (**Figure 2D and E**; **Shah et al., 2016**), consistent with the maintenance of general EC markers in culture.

Comparing the abundances of TF transcripts, **Figure 2F** shows that, relative to acutely isolated brain ECs, cultured brain ECs showed reductions in the abundances of transcripts for brain EC-enriched TFs *Lef1*, *Tcf7*, and *Zic3*, and increases in the abundances of transcripts for AP-1 family members *Fosl2* and *Jund* (**Figure 2F**). The down-regulation of transcripts coding for TFs such as *Lef1*, *Tcf7*, and *Zic3* that mediate beta-catenin signaling could be both a consequence and a cause of reduced signaling by this pathway.

Interestingly, transcripts coding for SP5, a direct target and downstream effector of beta-catenin signaling (**Weidinger et al., 2005**; **Fujimura et al., 2007**), are up-regulated in cultured ECs, whereas transcripts coding for SP1, a widely expressed TF that recognizes the same SP/KLF motif, are down-regulated (arrows in **Figure 2F**). Current evidence indicates that SP5 and its close homologue SP8 mediate many of the effects of Wnt3a signaling on neural and mesodermal development during mammalian gastrulation (**Dunty et al., 2014**; **Kennedy et al., 2016**). In HEK293T cells and human pluripotent stem cells, SP5 acts as a beta-catenin-inducible transcriptional repressor to inhibit the expression of genes that are activated by SP1 (**Fujimura et al., 2007**; **Huggins et al., 2017**), and Huggins and colleagues have proposed that SP5 acts in the termination phase of beta-catenin signaling to repress beta-catenin target genes. Pan-EC TF genes *Erg*, *Fli1*, and *Ets1* show minimal changes in expression in cultured brain ECs, consistent with the maintenance of general EC markers in culture.

## Stabilization of beta-catenin does not promote a BBB-like state in cultured CNS ECs

The data described thus far support the general inference from co-culture experiments with ECs and astrocytes, pericytes, and/or neurons that signals from the brain parenchyma are required to maintain the BBB state (**Helms et al., 2016**). More specifically, the data are consistent with mouse genetic experiments showing that a neuronal- and/or glial-derived beta-catenin signal is necessary for BBB development and maintenance in vivo, as summarized in the Introduction. To explore this idea in the context of our short-term culture system, we used a genetic strategy to artificially activate beta-catenin signaling in ECs prior to in vitro culture. The strategy relies on Cre-mediated excision of exon 3 of the beta-catenin gene (*Ctnnb1*), which encompasses the sites of phosphorylation that lead to beta-catenin ubiquitination and degradation (**Figure 3A**; **Harada et al., 1999**; the allele with *loxP* sites flanking exon 3 is referred to as *Ctnnb1*$^{flex3}$). In previous work, Cre-mediated recombination of *Ctnnb1*$^{flex3}$ in ECs was induced in vivo using a *Pdgfb-CreER* transgene and intraperitoneal injection of 4-hydroxytamoxifen (4HT), a procedure that reliably produces ~80% efficient recombination, as determined by the partial conversion of CVO and choroid plexus ECs from a BBB-deficient

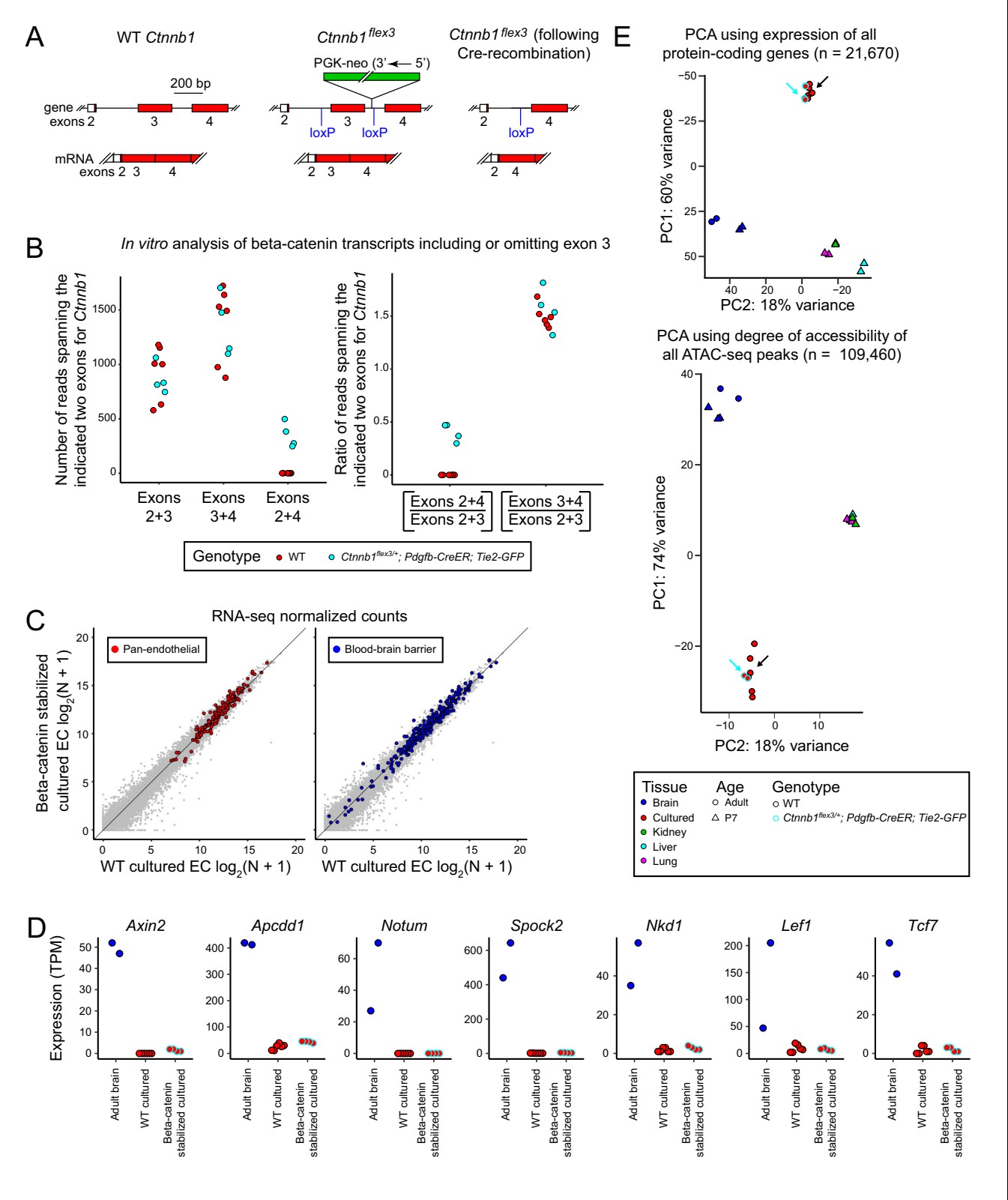

**Figure 3.** Stabilization of beta-catenin does not promote a BBB-like state in cultured CNS ECs as determined by RNA-seq and ATAC-seq. (**A**) Schematic outline of the genetic strategy for stabilizing beta-catenin (encoded by *Ctnnb1*). The *Ctnnb1flex3* allele contains *loxP* sites flanking exon 3. Cre-recombination produces an in-frame deletion of exon 3, and the resulting beta-catenin protein is stabilized and activates transcription in conjunction with LEF/TCF proteins. (**B**) Analysis of *Ctnnb1* transcripts that include or omit exon 3. The six replicates of wild-type (WT) cultured adult

*Figure 3 continued on next page*

*Figure 3 continued*

brain ECs produced no RNA-seq reads that join exons 2+4 whereas each of the four replicates of *Ctnnb1*<sup>flex3/+</sup>;*Pdgfb-CreER;Tie2-GFP* cultured adult brain ECs (in which exon 3 is deleted by Cre-mediated recombination) produced several hundred RNA-seq reads that join exons 2+4. (C) Scatter plots comparing cross-sample normalized RNA-seq read counts for protein-coding genes between beta-catenin stabilized and WT primary brain ECs cultured for 8 days. The left plot highlights (in red) EC-enriched genes that are expressed at similar levels in brain, liver, lung, and kidney EC subtypes. The right plot highlights (in blue) genes associated with the blood-brain barrier (BBB). Gene sets are as described for *Figure 1B*. Stabilizing beta-catenin does not increase the expression of BBB genes in cultured ECs. (D) Transcript abundances for seven genes from the 'Regulated by beta-catenin signaling' category. (E) PCA of protein-coding transcript abundances (RNA-seq; top) and PCA of ATAC-seq read density at all called ATAC-seq peaks (bottom) from independent biological replicates of WT and beta-catenin stabilized cultured adult brain ECs, acutely isolated adult and P7 brain ECs, and P7 liver, lung, and kidney ECs. The cyan-outlined red circles correspond to the beta-catenin stabilized cultured adult brain EC samples, and are indicated by cyan arrows. Black arrows point to cultured brain ECs without stabilized beta-catenin.

The online version of this article includes the following figure supplement(s) for figure 3:

**Figure supplement 1.** In vivo analysis of transcripts from FACS-purified pituitary ECs that include or omit *Ctnnb1* exon 3.
**Figure supplement 2.** WT and beta-catenin stabilized brain ECs in culture have nearly identical patterns of transcription and accessible chromatin.

to a BBB-proficient state (*Wang et al., 2019*). In the present work, we have used the same mouse line and the same protocol as used for the CVO and choroid plexus conversion experiments, with CNS ECs harvested from *Ctnnb1*<sup>flex3</sup>;*Pdgfb-CreER;Tie2-GFP* mice 10 weeks after 4HT treatment.

A comparison of RNA-seq read counts from brain ECs cultured from wild-type (WT) control (either C57BL/6J or *Ctnnb1*<sup>flex3/+</sup>;*Tie2-GFP*) injected with 100 ug 4HT at P10 and *Ctnnb1*<sup>flex3/+</sup>; *Pdgfb-CreER;Tie2-GFP* mice injected with 100 ug 4HT at P10 reveals reads spanning *Ctnnb1* exons 2+4 only in the genotype carrying the *Pdgfb-CreER* and *Ctnnb1*<sup>flex3</sup> alleles, implying that cultured brain ECs contain the exon 3 deleted transcripts that code for stabilized beta-catenin (*Figure 3A and B*). Similar ratios of *Ctnnb1* transcripts including or omitting exon 3 were observed in RNA-seq data sets obtained by *Wang et al. (2019)* from FACS-purified ECs from anterior and posterior pituitaries from control and *Ctnnb1*<sup>flex3/+</sup>;*Pdgfb-CreER;Tie2-GFP* mice (*Figure 3—figure supplement 1*).

Surprisingly, scatter plots comparing between-sample normalized RNA-seq read counts for WT versus beta-catenin stabilized cultured brain ECs show that expression of the exon 3 deleted allele of *Ctnnb1* had virtually no effect on the transcriptional profile of cultured brain ECs, and, in particular, had virtually no effect on either pan-EC transcripts or BBB transcripts (*Figure 3C*). Moreover, the expression of the six genes regulated by beta-catenin signaling that were shown in *Figure 1E*, and a seventh (*Tcf7*) were extremely low in beta-catenin stabilized cultured brain ECs (*Figure 3D*), implying a failure to activate beta-catenin signaling despite production of stabilized beta-catenin. PCA of transcript abundances for all protein-coding genes and PCA of all called ATAC-seq peaks also indicates that beta-catenin stabilized cultured brain ECs are virtually identical to their WT counterparts (*Figure 3E*, cyan and black arrows). The PCA analysis is consistent with visual inspection of the genomic alignments of RNA-seq and ATAC-seq reads (*Figure 3—figure supplement 2*) and with scatterplots of the abundances of transcripts coding for TJ proteins, solute carrier transporters, ABC transporters, and transcytosis proteins (*Figure 1—figure supplements 2* and *3*). The simplest interpretation of these findings is that under the conventional in vitro culture conditions employed here, CNS ECs become refractory to the effects of beta-catenin stabilization.

## Discussion

Organ-specific EC heterogeneity has been well-characterized at the morphological and functional levels, but the processes by which ECs are programmed to adopt organ-specific gene expression patterns remain largely unknown (*Aird, 2007*; *Potente and Mäkinen, 2017*). In the case of CNS vascular development, neural tissue induces a BBB/BRB gene expression program in ECs that invade from the surrounding perineural vascular plexus (*Feeney and Watterson, 1946*; *Stewart and Wiley, 1981*). As noted in the Introduction, gene knockout studies in mice have demonstrated that an essential component of this BBB/BRB instructive signal is beta-catenin signaling, induced by neuron- and/or glia-derived ligands (*Liebner et al., 2008*; *Stenman et al., 2008*; *Daneman et al., 2009*; *Wang et al., 2012*; *Zhou et al., 2014*; *Cho et al., 2017*; *Benz et al., 2019*; *Wang et al., 2019*). Two additional pieces of evidence that underscore the central role of beta-catenin signaling in CNS EC-

specific gene regulation are: (1) the presence in CNS ECs, but not in non-CNS ECs, of LEF1 and TCF7, members of the TCF/LEF family of transcription factors (TFs) that both mediate and are induced by the beta-catenin signal, and (2) enrichment of TCF/LEF binding motifs in brain EC-specific genomic regions with increased chromatin accessibility and reduced cytosine methylation, features associated with *cis*-regulatory elements (*Sabbagh et al., 2018*).

In short-term cultures of brain ECs, the present work reveals that the rapid reduction in the abundance of many BBB transcripts is correlated with reduced beta-catenin signaling. In contrast, the expression of generic EC genes, such as *Cd31/Pecam1*, is minimally affected. The reduction in beta-catenin signaling in culture is seen at both the transcriptional and chromatin accessibility levels. For example, multiple genes that are known beta-catenin targets and that are expressed in acutely isolated brain ECs are not detectably expressed or are expressed at extremely low levels in cultured brain ECs. Furthermore, while cultured brain ECs retain the expression of generic EC TF genes such as *Erg, Ets1,* and *Fli1*, the drastic reduction in expression of *Lef1* and *Tcf7*, which are both mediators and targets of beta-catenin signaling, likely contributes to the reduced response of cultured ECs to beta-catenin stabilization. One simple model for the difference between in vivo and in vitro patterns of gene expression is that under low beta-catenin signaling and no blood flow culture conditions, brain ECs utilize ETS family TFs in conjunction with AP1 and TEAD TFs to promote proliferation, whereas under the high beta-catenin signaling and normal blood flow conditions of the in vivo CNS, ETS TFs collaborate with SP/KLF, TCF/LEF, and ZIC factors to promote BBB gene expression. The lability of the BBB program in vitro is reminiscent of earlier experiments in which acute loss of EC beta-catenin signaling in vivo led to a loss of BBB markers and a loss of barrier integrity (*Liebner et al., 2008*; *Wang et al., 2012*).

We note that the comparison between in vivo and cultured CNS ECs could conceivably be affected by the presence of GFP expression in the in vivo ECs and/or by the inclusion of a 2–2.5 day regimen of puromycin as part of the in vitro isolation procedure. We think the former is unlikely because, in the *Tie2-GFP* line, the GFP is unfused to other proteins and the level of GFP accumulation in ECs is low. We think that the latter is unlikely because following puromycin treatment ECs were cultured for an additional 5–6 days before they were harvested. We also note that the experiments reported here were carried under only a single culture condition: 5% bovine serum with a standard mixture of EC growth factors (see Materials and methods). The conditions used here are typical for CNS EC cultures (*Helms et al., 2016*), and it would be interesting to determine the extent to which variations in percent serum and culture media can affect BBB-specific gene expression.

Based on the ability of stabilized beta-catenin to partially convert CVO and choroid plexus ECs to a BBB-like state (*Benz et al., 2019*; *Wang et al., 2019*), one might have expected that CNS ECs cultured from 4HT treated *Ctnnb1$^{flex3/+}$;Pdgfb-CreER* mice would show a similar BBB-like transcriptional and chromatin accessibility state. However, as the experiments in *Figure 3* illustrate, the presence of a constitutively active form of beta-catenin has virtually no effect on the gene expression program of short-term cultured CNS ECs, and, in particular, it is unable to maintain the expression of BBB genes. Previous work with cultured brain ECs and with immortalized CNS EC lines has shown that activating beta-catenin signaling with Wnt ligands, small molecule agonists, or by genetic means can enhance the expression of genes specific to the BBB (*Paolinelli et al., 2013*). While our data appear to be at odds with the data of *Paolinelli et al. (2013)*, we note that Paollinelli and colleagues observed relatively modest effects of beta-catenin stabilization, and they did not include acutely isolated CNS ECs in their comparisons, leaving open the possibility that the transcriptional effects they observed were minor compared to the differences between in vivo and in vitro transcriptional programs.

Based on our genome-wide transcriptional and chromatin accessibility analyses, we suggest that one or more additional signaling pathways gate the beta-catenin signal in CNS ECs and that the relevant signals for these other pathways are missing from the in vitro culture system. If correct, this model implies that during normal development ECs encounter additional signals in the CNS that render them receptive to the beta-catenin signal. Whether these hypothesized signals include laminar blood flow and/or neural-specific molecules remain to be determined.

Finally, the present study serves as a demonstration of the utility of applying genome-wide transcriptional and chromatin accessibility profiling to the characterization of ECs in culture. This combination of approaches has been used previously to distinguish tumor and immune cell subtypes (*Corces et al., 2018*; *Satpathy et al., 2019*), and it should be useful in defining the differentiation

state of CNS ECs grown under different culture or co-culture conditions or in the presence of candidate signaling molecules.

# Materials and methods

### Key resources table

| Reagent type (species) or resource | Designation | Source or reference | Identifiers | Additional information |
|---|---|---|---|---|
| Genetic reagent (*Mus musculus*) | *Ctnnb1*[flex3] | DOI: 10.1093/emboj/18.21.5931 | | |
| Genetic reagent (*M. musculus*) | *Tie2-Cre (Tek-Cre)* | The Jackson Laboratory | Stock No: 008863; RRID:IMSR_JAX:008863 | |
| Genetic reagent (*M. musculus*) | *Tie2-GFP (Tek-GFP)* | The Jackson Laboratory | Stock No: 003658; RRID:IMSR_JAX:003658 | |
| Genetic reagent (*M. musculus*) | *Pdgfb-CreER* | DOI: 10.1002/dvg.20367 | | |
| Peptide, recombinant protein | Tn5 transposase | Illumina | Cat no: FC-121–1030 | |
| Commercial assay or kit | Worthington Papain Dissociation Kit | Worthington Biochemical Corporation | Cat no: LK003160 | |
| Chemical compound, drug | 4-hydroxytamoxifen | Sigma-Aldrich | Cat no: H7904 | |
| Software, algorithm | Salmon | DOI: 10.1038/nmeth.4197 | | |
| Software, algorithm | deepTools | DOI: 10.1093/nar/gkw257 | RRID:SCR_016366 | |
| Software, algorithm | tximport | DOI: 10.12688/f1000research.7563.2 | RRID:SCR_016752 | |
| Software, algorithm | EBSeq | DOI: 10.1093/bioinformatics/btv193 | RRID:SCR_003526 | |
| Software, algorithm | MACS2 | DOI: 10.1186/gb-2008-9-9-r137 | | |
| Software, algorithm | DiffBind | DOI: 10.1038/nature10730 | RRID:SCR_012918 | |

## Mice

The following mouse alleles were used: *Ctnnb1*[flex3] (*Harada et al., 1999*); *Tie2-GFP* (also known as *Tek-GFP*; *Motoike et al., 2000*; JAX 003658); *Pdgfb-CreER* (*Claxton et al., 2008*); *Hprt*[LSL-GFP] (*Wu et al., 2014*); *Tie2-Cre* (also known as *Tek-Cre*; *Kisanuki et al., 2001*; JAX 008863); and C57BL/6J (JAX 000664). For EC culture experiments, the mice consisted of control WT mice (either C57BL/6J or *Ctnnb1*[flex3/+];*Tie2-GFP* that had received 100 ug 4HT at P10) and EC-specific beta-catenin stabilized mice (*Ctnnb1*[flex3/+];*Pdgfb-CreER*;*Tie2-GFP* that had received 100 ug 4HT at P10). To control for the possibility of sex-dependent differences, male mice were used for all experiments. All mice were housed and handled according to the approved Institutional Animal Care and Use Committee (IACUC) protocol MO16M367 of the Johns Hopkins Medical Institutions.

## Primary brain EC culture

Primary brain ECs were isolated and purified by either method A or method B, as described below. Both protocols take advantage of the specific expression of the multidrug efflux pump P-glycoprotein in brain ECs, which allows for positive selection using puromycin.

Method A was performed as previously described with slight modifications (*Welser-Alves et al., 2014*). Briefly, brains were harvested from six mice at two months of age, minced with a razor blade, and digested in 5 ml of papain solution (20 units papain per ml; 1 mM L-cysteine; 0.5 mM EDTA; 100 units of DNase per ml) for one hour and ten minutes at 37° Celsius. Following enzymatic digestion, the tissue was gently triturated with a 10 ml pipet to break up vascular fragments from the brain parenchyma. The suspension was then mixed with 7 ml of 30% BSA solution and centrifuged at 1120 x g for 10 min at room temperature. Pelleted vascular fragments were washed once with 2 ml of EC growth medium (ECGM; EGM-2 MV, CC-3202, Lonza, Basel, Switzerland; components: hEGF, hydrocortisone, 5% fetal bovine serum, VEGF, hFGF-B, R$^3$-IGF-1, ascorbic acid, and gentamicin and amphotericin-B replaced by penicillin and streptomycin), centrifuged at 300 x g for 5 min,

resuspended in 8 ml of ECGM, and then distributed evenly into four collagen type one coated wells of a six-well plate. Cells were incubated at 37° Celsius overnight. The following morning, the cells were washed seven times with DMEM/F12 medium (12500–062, Thermo Fisher Scientific) and then placed in fresh ECGM. In the early evening, the media was replaced with ECGM plus puromycin (P8833-10MG, Sigma-Aldrich; 4 ug/ml) to kill non-ECs. Puromycin treatment was stopped after 2.5 days. Cells were cultured for eight days before harvesting for RNA-seq and ATAC-seq. Biological replicates consisted of cells from independent wells. To assess the purity of cultures achieved using this isolation protocol, we cultured brain ECs from *Tie2-Cre;Hprt*$^{LSL-GFP}$ (*Wu et al., 2014*) mice expressing a nuclear-localized GFP in vascular ECs. Using the percentage of double GFP-positive and DAPI-positive nuclei as an assessment of EC purity, cultures exhibited >98% purity.

Method B was performed as previously described with slight modifications (*Assmann et al., 2017*). Briefly, brains from 4 to 6 mice were placed in 5 mL of Dulbecco's PBS (DPBS) in a Dounce tissue grinder and homogenized (30 strokes with pestle A, and 25 strokes with pestle B). Following homogenization, 7 mL of 30% BSA solution was mixed with the homogenate. The mixture was then centrifuged for 10 min at 1120 x g at room temperature. After removal of the floating myelin layer and the BSA solution, the pelleted vascular fragments were resuspended in 10 mL of DMEM/F12 solution containing 1 mg/mL of STEMxyme collagenase/dispase (Worthington Biochemical, LS004106) and 4 ug/mL of DNase. The fragments were digested for 1 hr at 37° Celsius and then centrifuged at 440 x g for 5 min at room temperature. The digestion media was removed, and the cells were washed with 10 mL of warm DPBS. After another centrifugation at 440 x g for 5 min at room temperature, the pelleted cells were resuspended in an appropriate volume of ECGM supplemented with puromycin at 8 ug/mL in order to plate the cells at roughly 0.1 brains per cm$^2$ of type one collagen-coated wells. Cells were incubated at 37° Celsius overnight, and the following day washed twice with DPBS. Fresh ECGM supplemented with puromycin was added, and the medium was changed the next day to puromycin-free ECGM.

Method A was used to generate RNA-seq WT and beta-catenin stabilized cultured brain EC replicates one and two as well as ATAC-seq WT cultured brain EC replicates one and two. Control cultured ECs processed by Method A were from C57BL/6J mice. Method B was used to generate RNA-seq WT cultured brain EC replicates three to six, RNA-seq beta-catenin stabilized cultured brain EC replicates three and four, ATAC-seq WT cultured brain EC replicates three to six, and ATAC-seq beta-catenin stabilized cultured brain EC replicates one and two. Control cultured ECs processed by Method B were from *Ctnnb1*$^{flex3/+}$;*Tie2-GFP* mice.

## Immunocytochemistry

ECs grown on coverslips were washed once with PBS and then fixed in 4% paraformaldehyde (PFA) for 15 min at room temperature. The following reagents were used: DAPI and rat anti-CD31 (1:300; 553370, BD BioSciences, San Jose, CA). Cells were incubated overnight with primary antibody diluted in 1x PBSTC (1x PBS + 1% Triton X-100 + 0.1 mM CaCl$_2$) + 10% normal goat serum (NGS). Incubation and washing steps were performed at 4°C. Cells were washed at least three times with 1x PBSTC, and subsequently incubated overnight with a secondary antibody plus DAPI diluted in 1x PBSTC +10% NGS. Because a primary antibody raised in rat was used, secondary antibodies were additionally incubated with 1% normal mouse serum (NMS) as a blocking agent. The next day, cells were washed at least three times with 1x PBSTC, and mounted on a slide using Fluoromount G (EM Sciences 17984–25). Cells were imaged using a Zeiss LSM700 confocal microscope and processed with ImageJ.

## 4HT preparation and administration

Solid 4HT (Sigma-Aldrich H7904) was dissolved at 20 mg/ml in ethanol by extensive vortexing. Sunflower seed oil (Sigma-Aldrich S5007) was added to dilute the 4HT to 2 mg/ml and aliquots were stored at −80°C. Thawed aliquots were diluted with Sunflower seed oil to a final concentration of 1 mg/ml 4HT. All injections were performed intraperitoneally.

## Acute isolation of adult brain ECs

Viable adult brain ECs were isolated from homozygous *Tie2-GFP* mice using the Worthington Papain Dissociation System (LK003160, Worthington Biochemical Corporation, Lakewood, NJ) and a MoFlo

XDP Sorter (Beckman Coulter, Brea, CA) as previously described (*Sabbagh et al., 2018*), with propidium iodide negative cells considered as viable. To remove myelin from the dissociated brain, the suspension was mixed with 30% BSA solution (1:1 vol) and centrifuged at 1120 x g for 10 min at room temperature.

## Sample processing for RNA-seq and ATAC-seq

For acutely isolated ECs, RNA was extracted from GFP-positive and GFP-negative cells that were FACS sorted directly into QIAGEN Buffer RLT Plus and then processed using the RNeasy Micro Plus kit (74034, QIAGEN, Venlo, Netherlands). For cultured ECs, RNA was extracted by adding QIAGEN Buffer RLT Plus directly into the well followed by extraction using the RNeasy Micro Plus kit. For ATAC-seq, ~50,000 GFP-positive FACS-sorted cells or trypsinized cultured ECs were gently centrifuged and then resuspended in ice-cold lysis buffer (0.25 M sucrose, 25 mM KCl, 5 mM MgCl$_2$, 20 mM Tricine-KOH, 0.1% Igepal CA-630) and immediately centrifuged at 500 x g for 10 min at 4°C to prepare nuclei. The resulting nuclear pellet was resuspended in a 50 ul reaction volume in Tn5 transposase and transposase reaction buffer (FC-121–1030, Illumina Inc, San Diego, CA), and the tagmentation reaction was incubated at 37°C for 30 min.

## Library preparation and sequencing

Libraries for RNA-seq and ATAC-seq were prepared as previously described (*Buenrostro et al., 2015*; *Sabbagh et al., 2018*). Adult brain EC RNA-seq replicates were single-end sequenced for 75 cycles on a NextSeq500 (Illumina). WT cultured brain EC RNA-seq libraries for replicates one, two, five, and six and beta-catenin stabilized brain EC RNA-seq libraries for all four replicates were paired-end sequenced for 36 cycles on a NextSeq500. WT cultured brain EC RNA-seq libraries for replicates three and four were prepared by Omega Bioservices (Georgia) and paired-end sequenced for 150 cycles. Tagmented DNA was purified using QIAGEN MinElute Gel Extraction kit (28604, Qiagen). ATAC-seq libraries were PCR amplified for 11 cycles. Agencourt AMPure XP beads (A63880, Beckman Coulter) were used to purify ATAC-seq libraries, which were then paired-end sequenced for 36 cycles on a NextSeq500 (Illumina). Sequencing libraries that contained overrepresented adaptor sequences were trimmed using Trim Galore (https://github.com/FelixKrueger/TrimGalore).

## Data analysis

Most data analyses were performed as previously described (*Sabbagh et al., 2018*). For basic data processing, exploration, and visualization, we used deepTools (*Ramírez et al., 2016*), the tidyverse collection of R packages (*Wickham, 2017*), ggplot2 (*Wickham, 2009*), and pheatmap (*Kolde, 2015*).

## RNA-seq data analysis

Salmon version 0.14.0 (*Patro et al., 2017*) was used to quantify expression of transcripts from RNA-seq experiments using GENCODE release M21 (`salmon quant -l A –validateMappings –mimicStrictBT2 –rangeFactorizationBins 4 –incompatPrior 0.0 –useVBOpt –seq-Bias –gcBias –posBias –biasSpeedSamp 10`). To visualize RNA-seq data on an IGV browser (*Robinson et al., 2011*; *Thorvaldsdóttir et al., 2013*), HISAT2 version 2.1.0 (*Kim et al., 2015*) was used to generate alignment bam files (`hisat2 –dta`) and then deepTools was used to generate bigwig files (`bamCoverage -bs 1 –normalizeUsing CPM`). To convert transcript-level abundances to the gene-level for further downstream analyses, we used tximport version 1.12.3 (*Soneson et al., 2015*).

Differentially expressed genes were identified using EBSeq version 1.24.0 (*Leng and Kendziorski, 2015*). A gene was considered to be differentially expressed if it met the following criteria: (1) a minimum two-fold enrichment between samples; (2) a posterior probability of differential expression (PPDE) greater than or equal to 0.95 [PPDE = (1 - false discovery rate)], that is an FDR < 0.05; and (3) a TPM value greater than or equal to five in both biological replicates. To filter out background transcripts from surrounding parenchymal cells, a set of EC-enriched transcripts was determined for each tissue by comparing RNA-seq data from GFP-positive and GFP-negative sorted cells. A gene was considered EC-enriched if it met the following three criteria: (1) a minimum two-fold enrichment in GFP-positive compared to GFP-negative samples, (2) a PPDE greater than or equal to 0.95, and

(3) relative expression greater than or equal to five transcripts per million (TPM) in both biological replicates. Principal component analysis was performed on variance stabilizing transformed data using the DESeq2 *vst* and *plotPCA* function (*Love et al., 2014*).

As described under 'Library preparation and sequencing', RNA samples were sequenced with paired-end 150 nt, single-end 75 nt, and paired-end 36 nt Illumina protocols. To test whether these different RNA-seq protocols affected genome alignment – and, therefore, differential expression analysis – we used the FASTX-Toolkit to trim the raw reads from one of the paired-end 150 nt samples to create two single-end 75 nt datasets, and a paired-end 36 nt data set (`fastx_trimmer -Q 33 l -z`). We then compared Salmon (version 0.14.0) quantification of gene expression from (1) the original 150 nt paired-end reads, (2) the trimmed 75 nt reads from one mate from the original pair, (2) the trimmed 75 nt reads of the other mate from each original pair, and (4) the trimmed 36 nt paired-end reads. Correlation coefficients for each of the four pair-wise comparisons were calculated, and the results show all four data sets were highly correlated with Pearson's r ranging from 0.994 to 0.998.

## ATAC-seq data analysis

ATAC-seq data were aligned to the GRCm38 genome using HISAT2 version 2.1.0 (`hisat2 -t -X 2000 -no-mixed -no-discordant`) and then duplicate reads were removed (*picard MarkDuplicates*). Peaks were called using MACS2 version 2.1.2 (`callpeak -f BAM -nomodel -keep-dup all -shift −100 -extsize 200`) (*Zhang et al., 2008*). Peaks were then filtered for fold-change >2 and -log(qvalue)>2. deepTools was used to visualize ATAC-seq peaks on the browser (`bamCoverage -bs 1 -extendReads -ignoreDuplicates -normalizeUsing CPM`). To identify differential ATAC-seq peaks between acutely isolated and cultured adult brain ECs, DiffBind version 2.12.0 was used (*Stark and Brown, 2011*; *Ross-Innes et al., 2012*) with EdgeR (*Robinson et al., 2010*). For the comparison, DiffBind was used to develop a set of consensus peaks between replicates using the requirement that peaks must be in a minimum of two replicates (minOverlap = 2). To retrieve a set of high-confidence, cell type-enriched peaks, we filtered for peaks with an absolute fold difference >2 and FDR < 0.05. Principal component analysis was performed on 'regularized' log-transformed data using the DESeq2 *rlog* and *plotPCA* function. To identify transcription factor DNA binding motifs enriched in ATAC-seq peaks, the HOMER suite of tools was used for motif discovery (*Heinz et al., 2010*), in particular, `findMotifsGenome.pl -size given`. `AnnotatePeaks.pl -m -size 1000 -hist` 5 was used to generate histograms of enriched motifs.

# Acknowledgements

The authors thank Yanshu Wang for providing the 4HT treated mice and assessing the culture potential of FACS-sorted brain ECs, and Hao Zhang for FACS sorting the brain ECs. Supported by the Howard Hughes Medical Institute, the National Eye Institute (NIH) (R01EY018637 to JN), and the National Institute of Child Health and Development (F30HD088023 to MS).

# Additional information

### Competing interests

Jeremy Nathans: Reviewing Editor, eLife, and is a paid consultant for Cerevance and Atengen. The other author declares that no competing interests exist.

### Funding

| Funder | Grant reference number | Author |
| --- | --- | --- |
| Howard Hughes Medical Institute | | Jeremy Nathans |
| National Eye Institute | R01EY018637 | Jeremy Nathans |
| Eunice Kennedy Shriver National Institute of Child Health and Human Development | F30HD088023 | Mark F Sabbagh |

The funders had no role in study design, data collection and interpretation, or the decision to submit the work for publication.

### Author contributions
Mark F Sabbagh, Conceptualization, Data curation, Formal analysis, Funding acquisition, Investigation, Visualization; Jeremy Nathans, Conceptualization, Resources, Formal analysis, Supervision

### Author ORCIDs
Mark F Sabbagh (iD) http://orcid.org/0000-0003-1996-5251
Jeremy Nathans (iD) https://orcid.org/0000-0001-8106-5460

### Ethics
Animal experimentation: All mice were housed and handled according to the approved Institutional Animal Care and Use Committee (IACUC) protocol MO16M367 of the Johns Hopkins Medical Institutions.

### Decision letter and Author response
Decision letter https://doi.org/10.7554/eLife.51276.sa1
Author response https://doi.org/10.7554/eLife.51276.sa2

## Additional files
### Supplementary files
• Supplementary file 1. Gene expression data. (**A**) EC-enriched transcripts for adult brain. PPDE, posterior probability of differential expression (1-FDR). (**B**) Differentially expressed genes in acutely isolated adult brain ECs when compared to cultured brain ECs. (**C**) The intersection of genes in (A) and (B). EC-enriched differentially expressed genes in acutely isolated adult brain ECs. (**D**) Differentially expressed genes in cultured brain ECs when compared to acutely isolated adult brain ECs. (**E**) All EC-enriched genes. (**F**) Pan-endothelial cell genes. (**G**) Tissue-specific EC genes. (**H**) Blood-brain barrier genes. (**I**) Blood-brain barrier genes with reduced expression in cultured brain ECs.

• Supplementary file 2. Accessible chromatin peaks. (**A**) Differential ATAC-seq peaks in acutely isolated adult brain ECs when compared to cultured brain ECs. (**B**) Differential ATAC-seq peaks in cultured brain ECs when compared to acutely isolated adult brain ECs.

• Transparent reporting form

### Data availability
Sequencing data have been deposited in GEO under accession code GSE118731.

The following dataset was generated:

| Author(s) | Year | Dataset title | Dataset URL | Database and Identifier |
|---|---|---|---|---|
| Sabbagh MF, Nathans J | 2019 | A genome-wide view of the de-differentiation of central nervous system endothelial cells in culture | https://www.ncbi.nlm.nih.gov/geo/query/acc.cgi?acc=GSE118731 | NCBI Gene Expression Omnibus, GSE118731 |

The following previously published datasets were used:

| Author(s) | Year | Dataset title | Dataset URL | Database and Identifier |
|---|---|---|---|---|
| Sabbagh MF, Heng J, Luo C, Castanon RG, Nery JR, Rattner A, Goff LA, Ecker JR, Nathans J | 2018 | Transcriptional and Epigenomic Landscapes of CNS and non-CNS Vascular Endothelial Cells | https://www.ncbi.nlm.nih.gov/geo/query/acc.cgi?acc=GSE111839 | NCBI Gene Expression Omnibus, GSE111839 |
| Wang Y, Sabbagh | 2019 | The role of beta-catenin signaling | https://www.ncbi.nlm. | NCBI Gene |

MF, Gu X, Rattner A, Williams J, Nathans J | in regulating barrier vs. non-barrier gene expression programs in circumventricular organ and ocular vasculatures | nih.gov/geo/query/acc. cgi?acc=GSE122117 | Expression Omnibus, GSE111839

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
