## [Decision Letter]

**Acceptance summary:**

We believe that this Research Advance describing epigenetic and transcriptional changes to vascular endothelial cells isolated from the central nervous system and cultured in vitro adds to the original study of freshly isolated CNS endothelial cells (Sabbagh et al., 2018). It is also interesting that forced expression of b-catenin that partially restores CNS patterns in vivo does not appear to be effective in vitro.

**Decision letter after peer review:**

Thank you for submitting your article "A genome-wide view of the de-differentiation of central nervous system endothelial cells in culture" for consideration by *eLife*. Your article has been reviewed by two peer reviewers, and the evaluation has been overseen by a Reviewing Editor and Didier Stainier as the Senior Editor. The following individuals involved in review of your submission have agreed to reveal their identity: Audrey Cleuren (Reviewer #1).

The reviewers have discussed the reviews with one another and the Reviewing Editor has drafted this decision to help you prepare a revised submission.

Summary:

This Research Advance compares the transcriptional and epigenetic landscapes of freshly isolated brain ECs versus cultured brain ECs. Data indicate loss of BBB-associated transcripts in brain ECs cultured in vitro, coincident with downregulation of the β-catenin signaling pathway. ECs with artificially activated β-catenin signaling prior to culturing were evaluated, hypothesizing that this would result in a better representation of the in vivo BBB EC phenotype. While stabilized β-catenin transcript levels are present in vitro, RNA-seq and ATAC-seq data show virtually no effect on profiles from wild-type versus mutant mice. Overall, the data presented illustrate the phenotypic drift brain ECs undergo during in vitro culture, which is particularly relevant for the evaluation and extrapolation of in vitro studies.

Essential revisions:

To bolster the presented results and increase the rigor of the study, two follow up experiments and one additional comparison and clarification are indicated:

1) The acutely isolated EC are homozygous for Tie2-GFP, while the cultured EC are isolated from WT mice and puromycin selection. A formal comparison of GFP+ sorted cells and EC directly after puromycin selection would help address differences due to isolation protocol vs. culture. This would also address potential differential selection of ECs due to use of different concentrations of puromycin in different experiments.

2) The lack of induction in Wnt responsive genes suggest that either some signals present in the CNS are missing, or alternatively that signals derived from the serum present in the EC medium may block function of activated β-catenin. TGF β signaling is highly induced by serum. Therefore, performing these experiments under culture conditions with very low serum (0.5%) in the absence and presence of TGFbeta inhibitors would better test if Wnt downstream targets are induced in culture.

3) In vitro experiments performed with brain ECs originating from Ctnnb1flex3/+ mice do not show substantial differences in transcriptional profiles as compared to ECs derived from wild-type mice, whereas previous work using these mice from Wang et al. (*eLife* 2019;8:e43257) demonstrated partial conversion from permeable ECs to a barrier-type state in vivo. The current study shows that after culturing, transcripts representing the stabilized β-catenin form are present, although appear to make up <50% of all transcripts (Figure 1—figure supplement 1B). It would be of interest to compare in vitro data to the previous in vivo data in Wang et al., particularly to see if there are more stable transcripts present in vivo and assess possibility of selection bias towards the wild-type (non-stabilized) allele when performing the EC isolation and culture.

4) It is not always clear which data are new and which represent reanalysis of data already reported in the authors' previous paper on which this report is based. For example, are the data from the acutely isolated brain ECs the same data already reported in the original manuscript? If so, this should be directly acknowledged/referenced in the current report.

---

## [Author Response]

Essential revisions:To bolster the presented results and increase the rigor of the study, two follow up experiments and one additional comparison and clarification are indicated:1) The acutely isolated EC are homozygous for Tie2-GFP, while the cultured EC are isolated from WT mice and puromycin selection. A formal comparison of GFP+ sorted cells and EC directly after puromycin selection would help address differences due to isolation protocol vs. culture. This would also address potential differential selection of ECs due to use of different concentrations of puromycin in different experiments.

With respect to the first sentence, while we have not formally ruled out an effect of GFP expression from Tie2-GFP on endothelial cell (EC) properties, we note that the level of GFP expression in the Tie2-GFP line is quite low; the GFP fluorescence is sufficient for detection by FACS, but to visualize the GFP signal by confocal microscopy in tissue sections we must use anti-GFP immunostaining (unlike other mouse lines such as mTmG, that have strong GFP or tdTomato expression). Moreover, the GFP in the Tie2-GFP line is unmodified so that it accumulates diffusely in the cell. Unlike histone-GFP fusions, free monomeric GFP is relatively benign. For these two reasons we think it highly unlikely that the presence of GFP in the Tie2-GFP line is altering the gene expression pattern of CNS ECs. With respect to the third sentence (“different concentrations of puromycin”), we think it is a strength of the present study that we explored minor variations on the puromycin protocol (one with 4 ug/ml puromycin and the other with 8 ug/ml puromycin) and obtained very similar results.

The goal of the present study was to assess the effect of a brief period of in vitro culture on CNS EC gene expression and chromatin state using the standard protocol for CNS EC isolation. The 8 days between cell isolation and harvesting for RNAseq and ATACseq is about as short an interval as is practical because one needs 2 days of puromycin treatment to kill non-endothelial cells (ECs) and a few more days to completely remove dying cells and obtain sufficient numbers of brain ECs to do the “omics” analyses. The yield of vascular fragments, from which the ECs emerge, is quite low – typically the vascular fragments from an entire adult mouse brain are sufficient for only one well of a 6-well plate and generate several tens of thousands viable ECs.

In response to this comment, we have tried purifying and culturing brain ECs using FACS sorting of GFP+ cells from the Tie2-GFP line (and thereby omitting the puromycin treatment), but almost no ECs obtained by this method survive in culture, most likely because ECs are highly sensitive to the mechanical trauma associated with FACS. For more than a decade, nearly all laboratories that have cultured acutely isolated brain ECs have used the protocols that we used, i.e. starting with vascular fragments purified by low speed centrifugation followed by puromycin selection to eliminate non-ECs.

2) The lack of induction in Wnt responsive genes suggest that either some signals present in the CNS are missing, or alternatively that signals derived from the serum present in the EC medium may block function of activated β-catenin. TGF β signaling is highly induced by serum. Therefore, performing these experiments under culture conditions with very low serum (0.5%) in the absence and presence of TGFbeta inhibitors would better test if Wnt downstream targets are induced in culture.

The culture conditions that we have used, and, in particular, our use of 5% serum and EC supplements, are representative of the standard culture conditions for CNS ECs in experiments aimed at studying BBB properties in vitro. As evidence of this statement, we have collated the% serum used in CNS EC cultures as reported in ten papers chosen at random from the review by Helms et al., 2016, which summarizes the present state of CNS EC culture conditions that are used to promote BBB properties. The ten references (including, in some cases, the original reference with the detailed protocol) and% serum are listed below. The% serum for EC culture ranges from 2% to 20%, with the majority using 10% or 20% serum.

The goal of the present study is not to explore variations in culture conditions that might make the cultured ECs more or less BBB-like, but to make a detailed and rigorous genome-wide assessment of the state of gene expression under the conditions for CNS EC culture that are the standard in the field. With regard to investigating the effect of lowering serum from 10% to 0.5%, as suggested above, this might prove interesting but it also may not be physiologic, since ECs in vivo are bathed (on one side) in 100% serum. In our view, a mechanistic analysis of the TGF-β signaling pathway (or other pathways that potentially intersect with the Wnt pathway) as beyond the scope of the present study.

11-10-2019 table of serum% for EC culture systems

Reference numbers are from the following review:

Helms HC, Abbott NJ, Burek M, Cecchelli R, Couraud PO, Deli MA, Förster C, Galla HJ, Romero IA, Shusta EV, Stebbins MJ, Vandenhaute E, Weksler B, Brodin B.

in vitro models of the blood-brain barrier: An overview of commonly used brain endothelial cell culture models and guidelines for their use. J Cereb Blood Flow Metab. 2016 May;36(5):862-90.

20% bovine serum (ref 96)

N. Perrière, P. Demeuse, E. Garcia, A. Regina, M. Debray, J.P. Andreux, P. Couvreur, J.M. Scherrmann, J. Temsamani, P.O. Couraud, M.A. Deli, F. Roux. Puromycin-based purification of rat brain capillary endothelial cell cultures. Effect on the expression of blood–brain barrier-specific properties.J. Neurochem., 93 (2005), pp. 279-289

Perrière N, Yousif S, Cazaubon S, Chaverot N, Bourasset F, Cisternino S, Declèves X, Hori S, Terasaki T, Deli M, Scherrmann JM, Temsamani J, Roux F, Couraud PO.

A functional in vitro model of rat blood-brain barrier for molecular analysis of efflux transporters. Brain Res. 2007 May 30;1150:1-13.

10% serum (ref 98)

Nakagawa S, Deli MA, Kawaguchi H, Shimizudani T, Shimono T, Kittel A, Tanaka K, Niwa M. A new blood-brain barrier model using primary rat brain endothelial cells, pericytes and astrocytes.Neurochem Int. 2009 Mar-Apr;54(3-4):253-63.

10% FBS (ref 99)

Dohgu S, Takata F, Yamauchi A, Nakagawa S, Egawa T, Naito M, Tsuruo T, Sawada Y, Niwa M, Kataoka Y. Brain pericytes contribute to the induction and up-regulation of blood-brain barrier functions through transforming growth factor-β production. Brain Res. 2005 Mar 21;1038(2):208-15.

2.5% FBS (ref 103)

Hatherell K1, Couraud PO, Romero IA, Weksler B, Pilkington GJ. Development of a three-dimensional, all-human in vitro model of the blood-brain barrier using mono-, co-, and tri-cultivation Transwell models. J Neurosci Methods. 2011 Aug 15;199(2):223-9.

Weksler BB1, Subileau EA, Perrière N, Charneau P, Holloway K, Leveque M, Tricoire-Leignel H, Nicotra A, Bourdoulous S, Turowski P, Male DK, Roux F, Greenwood J, Romero IA, Couraud PO. Blood-brain barrier-specific properties of a human adult brain endothelial cell line. FASEB J. 2005 Nov;19(13):1872-4.

20% FBS (ref 109)

Coisne C, Dehouck L, Faveeuw C, Delplace Y, Miller F, Landry C, Morissette C, Fenart L, Cecchelli R, Tremblay P, Dehouck B. Mouse syngenic in vitro blood-brain barrier model: a new tool to examine inflammatory events in cerebral endothelium. Lab Invest. 2005 Jun;85(6):734-46.

2% FCS (ref 110)

Förster C, Silwedel C, Golenhofen N, Burek M, Kietz S, Mankertz J, Drenckhahn D. Occludin as direct target for glucocorticoid-induced improvement of blood-brain barrier properties in a murine in vitro system. J Physiol. 2005 Jun 1;565(Pt 2):475-86.

10% FBS (ref 111)

Bernas MJ, Cardoso FL, Daley SK, Weinand ME, Campos AR, Ferreira AJ, Hoying JB, Witte MH, Brites D, Persidsky Y, Ramirez SH, Brito MA. Establishment of primary cultures of human brain microvascular endothelial cells to provide an in vitro cellular model of the blood-brain barrier. Nat Protoc. 2010 Jul;5(7):1265-72.

2.5% FBS (cells grown to confluence); then switched to 0.25% FBS for acute testing (ref 114)

Weksler BB, Subileau EA, Perrière N, Charneau P, Holloway K, Leveque M, Tricoire-Leignel H, Nicotra A, Bourdoulous S, Turowski P, Male DK, Roux F, Greenwood J, Romero IA, Couraud PO. Blood-brain barrier-specific properties of a human adult brain endothelial cell line. FASEB J. 2005 Nov;19(13):1872-4.

20% FBS (cells grown to confluence); then switched to serum-free for acute testing (ref 120)

Weidenfeller C, Schrot S, Zozulya A, Galla HJ. Murine brain capillary endothelial cells exhibit improved barrier properties under the influence of hydrocortisone. Brain Res. 2005 Aug 16;1053(1-2):162-74.

10% FBS (ref 123)

Omidi Y, Campbell L, Barar J, Connell D, Akhtar S, Gumbleton M. Evaluation of the immortalised mouse brain capillary endothelial cell line, b.End3, as an in vitro blood-brain barrier model for drug uptake and transport studies. Brain Res. 2003 Nov 14;990(1-2):95-112.

3) In vitro experiments performed with brain ECs originating from Ctnnb1flex3/+ mice do not show substantial differences in transcriptional profiles as compared to ECs derived from wild-type mice, whereas previous work using these mice from Wang et al. (eLife 2019;8:e43257) demonstrated partial conversion from permeable ECs to a barrier-type state in vivo. The current study shows that after culturing, transcripts representing the stabilized β-catenin form are present, although appear to make up <50% of all transcripts (Figure 1—figure supplement 1B). It would be of interest to compare in vitro data to the previous in vivo data in Wang et al., particularly to see if there are more stable transcripts present in vivo and assess possibility of selection bias towards the wild-type (non-stabilized) allele when performing the EC isolation and culture.

Just to clarify: the statement above that “Wang et al. (*eLife* 2019;8:e43257) demonstrated partial conversion from permeable ECs to a barrier-type state in vivo” refers specifically to vasculature in the circumventricular organs (CVOs) and the choroid plexus, those exceptional CNS vascular beds with highly permeable capillaries. Here we are studying ECs from the conventional brain vasculature, which has a BBB. In ECs that comprise the BBB+ brain vasculature, β-catenin is already stabilized to some extent by Wnt signaling, and further stabilization of β-catenin in vivo (by deleting exon 3) has only a minimal effect on gene expression, as demonstrated in Wang et al., 2019.

Comparing the fraction of Ctnnb1 transcripts from the in vivo experiments that have or do not have exon 3 is a good idea. We have now quantified the abundances of Ctnnb1 transcripts from the Wang et al., 2019, in vivo RNAseq data sets, representing FACS purified anterior and posterior pituitary ECs with or without Pdgfb-CreER mediated excision of Ctnnb1 exon 3 from the floxed allele. These data are now plotted in a new Figure 3—figure supplement 2. These data, like the data from the cultured ECs in the present study, show RNAseq reads spanning Ctnnb1 exons 2+4 (i.e. missing exon 3) only in Pdgfb-CreER;Ctnnb1flex3 mice that were treated with 4-hydroxytamoxifen (4HT). The in vivo data from Wang et al., 2019 show that, on average, ~25% of Ctnnb1 transcripts in the FACS sorted pituitary ECs are missing exon 3, as determined from the ratio of RNAseq reads spanning exons 2+4 compared to exons 2+3. With an estimated ~80% efficiency of CreER-mediated excision (as judged by the fraction of ECs that convert their immunostaining patterns) and only one of the two alleles carrying a floxed exon 3, we would expect ~40% of Ctnnb1 transcripts to be missing exon 3. In the present study, the cultured ECs obtained from the same mouse line, and treated in the same manner with 4HT, showed that, on average, ~40% of Ctnnb1 transcripts are missing exon 3 (Figure 3B). we note that the in vivo data from purified pituitary ECs shows more variability, most likely reflecting limited read coverage in that region of the Ctnnb1 mRNA. Our assessment is that, within experimental error, the in vivo (~25%) and in vitro (~40%) data are similar. This is now noted in the Results section.

4) It is not always clear which data are new and which represent reanalysis of data already reported in the authors' previous paper on which this report is based. For example, are the data from the acutely isolated brain ECs the same data already reported in the original manuscript? If so, this should be directly acknowledged/referenced in the current report.

Thank you for alerting us to this issue, which we have now clarified in the text. In this manuscript, adult brain ECs refers to combined data from adult cerebrum and cerebellum. The following data is new: ATACseq and RNAseq on ECs in culture and on ECs acutely isolated from adult cerebrum. The ATACseq and RNAseq from adult cerebellum were reported in Wang et al., 2019 and the P7 brain, liver, lung, and kidney ATACseq and RNAseq data were described in Sabbagh et al., 2018. The P7 brain EC single-cell RNA-seq was described in Sabbagh et al., 2018. This is now clarified in the Results and Materials and methods sections.